# Test-trace-isolate-quarantine (TTIQ) intervention strategies after symptomatic COVID-19 case identification

**Peter Ashcroft**\*, **Sonja Lehtinen**, **Sebastian Bonhoeffer**\*

Institute of Integrative Biology, ETH Zurich, Zürich, Switzerland

\* peter.ashcroft@env.ethz.ch (PA); seb@env.ethz.ch (SB)

**Data Availability Statement:** All code and data are publicly available on GitHub (http://github.com/ashcroftp/COVID-TTIQ).

**Funding:** This study was funded by the Swiss National Science Foundation (grant no.

## Abstract

The test-trace-isolate-quarantine (TTIQ) strategy, where confirmed-positive pathogen carriers are isolated from the community and their recent close contacts are identified and preemptively quarantined, is used to break chains of transmission during a disease outbreak. The protocol is frequently followed after an individual presents with disease symptoms, at which point they will be tested for the pathogen. This TTIQ strategy, along with hygiene and social distancing measures, make up the non-pharmaceutical interventions that are utilised to suppress the ongoing COVID-19 pandemic. Here we develop a tractable mathematical model of disease transmission and the TTIQ intervention to quantify how the probability of detecting and isolating a case following symptom onset, the fraction of contacts that are identified and quarantined, and the delays inherent to these processes impact epidemic growth. In the model, the timing of disease transmission and symptom onset, as well as the frequency of asymptomatic cases, is based on empirical distributions of SARS-CoV-2 infection dynamics, while the isolation of confirmed cases and quarantine of their contacts is implemented by truncating their respective infectious periods. We find that a successful TTIQ strategy requires intensive testing: the majority of transmission is prevented by isolating symptomatic individuals and doing so in a short amount of time. Despite the lesser impact, additional contact tracing and quarantine increases the parameter space in which an epidemic is controllable and is necessary to control epidemics with a high reproductive number. TTIQ could remain an important intervention for the foreseeable future of the COVID-19 pandemic due to slow vaccine rollout and highly-transmissible variants with the potential for vaccine escape. Our results can be used to assess how TTIQ can be improved and optimised, and the methodology represents an improvement over previous quantification methods that is applicable to future epidemic scenarios.

## Introduction

Individuals who are confirmed as infected with severe acute respiratory syndrome coronavirus 2 (SARS-CoV-2) are isolated from the community to prevent further transmission. Individuals who have been in recent close contact with an infected individual have an increased risk of being infected themselves. By identifying the potentially-infected contacts through contact

310030B_176401, awarded to SB). The funders had no role in study design, data collection and analysis, decision to publish, or preparation of the manuscript.

**Competing interests:** The authors have declared that no competing interests exist.

tracing and eventually quarantining them, transmission chains can be broken. Thus contact tracing is an essential public health tool for controlling epidemics [1]. The strategy of testing to identify infected cases, isolating them to prevent further transmission, and tracing & quarantining their recent close contacts is known as test-trace-isolate-quarantine (TTIQ) [2]. This strategy is a key non-pharmaceutical intervention which is used globally to control the ongoing COVID-19 pandemic [3].

Testing typically occurs once an individual develops symptoms indicative of coronavirus disease 2019 (COVID-19). As presymptomatic transmission makes up approximately 40% of total onward transmission from eventually-symptomatic infecteds [4–6], it would be possible for the number of secondary infections to be more than halved if infected individuals were isolated from the community at the time of symptom onset. However, as testing follows from symptoms in this scenario, the testing & isolating strategy without subsequent contact tracing & quarantine is unlikely to capture persistently-asymptomatic infections which make up around 20% of all infecteds [7], and thus isolating 100% of infecteds at symptom onset would not be possible.

Contact tracing & quarantine have the potential to be effective interventions against the spread of COVID-19 because of the high frequency of presymptomatic and asymptomatic transmission from recently-infected individuals [8]. Potentially-infected contacts can be identified and quarantined before they would be isolated as a result of developing symptoms and/or receiving a positive test result, such that their onward transmission is reduced. This is exemplified during super-spreader events [9–11] where large numbers of potentially-infected contacts can be quarantined to prevent widespread community transmission. Tracing & quarantine do not depend on symptom development, hence this strategy is capable of reducing onward transmission even from asymptomatically-infected individuals.

TTIQ strategies are not perfect: each stage in the process is subject to delays and uncertainties and it would be impossible to prevent all transmission through TTIQ alone [3, 12–16]. Here we will systematically assess how the probability of identifying and isolating a symptomatic case, the fraction of contacts identified and quarantined, and the delays that are inherent to these processes impact disease transmission under TTIQ interventions. From this quantification we can determine how TTIQ can be improved and optimised. For example, in the presence of widespread community transmission when contact tracers may be overwhelmed by the volume of cases, it is important to optimise the available resources (e.g. the person hours of the contact tracers) to minimise onward transmission.

Previous studies employing different approaches have begun to address these questions. Ferretti et al. [12] concluded that minimal delay between index case identification and quarantine of secondary contacts (as achieved with widespread digital contact tracing) is necessary to reduce the effective reproduction number below one and to bring an outbreak under control. Kretzschmar et al. [13] also predominantly focussed on digital contact tracing based on mobile applications, confirming the conclusions of Ferretti et al. [12] that minimising delays is the key to a successful TTIQ intervention. Through simulations based on real-world contact networks, Kucharski et al. [3] concluded that > 75% of contacts have to be quarantined (through manual contact tracing or digital app-based tracing) to reduce the effective reproduction number below one. Finally, Grantz et al. [17] employed a discrete-time Markov chain model of transmission, isolation, and contact tracing to conclude that effective TTIQ interventions need to be strong in the "test" component, as case detection underlies all other TTIQ components.

In this paper we build on our previous modelling work in which we have quantified the impact of quarantine duration and highlighted the optimal use of test-and-release strategies [16]. With this mathematical framework, which uses the empirically-observed distributions of

transmission timing from Ferretti et al. [6] to determine when infections occur, we systemati-
cally quantify the effective reproduction number in the presence of TTIQ.

Although a global roll-out of vaccines against SARS-CoV-2 is under way, non-pharmaceuti-
cal interventions are likely to remain crucial to epidemic control for the foreseeable future [18,
19]. Furthermore, we are still at risk from new variants which are highly-transmissible and/or
less impeded by current vaccines and acquired immunity. Our framework is flexible enough to
quantify the limitations of TTIQ in these scenarios, as well as provide insight about how these
interventions can be optimised.

## Materials and methods

### Transmission model

Our transmission model is based on a branching process that starts with a single individual
who is infected with SARS-CoV-2. The branching process model assumes discrete generations
of transmission and an infinite population size, such that the expected number of secondary
infections per infected is the same across generations. We therefore do not explicitly include
the depletion of susceptibles due to death or acquired immunity during epidemic spread and/
or vaccination campaigns. The initial infected individual could be persistently asymptomatic
(which make up a fraction $a$ of infections), otherwise they are classed as symptomatic $(1 - a)$.
To be clear, presymptomatic individuals who will go on to develop symptoms are included in
the symptomatic fraction. Buitrago-Garcia et al. [7] have estimated $a \approx 20\%$ based on a meta-
analysis of 79 studies.

The timing of onward infections in the model is determined by empirically-observed distri-
butions of transmission dynamics from Ferretti et al. [6]. These distributions are: the genera-
tion time distribution (describing the time interval between the infection of an index case and
secondary case); the infectivity profile (describing the time interval between the onset of symp-
toms in the index case and infection of the secondary case); and the incubation period distri-
bution (describing the time between the infection of an individual and the onset of their
symptoms). These distributions (shown in Fig I in S1 Appendix) are based on large sets of
transmission pairs and minimal assumptions about the relationship between infectiousness
and symptoms, which would otherwise push the variance of the resulting generation time dis-
tribution towards its upper or lower extremes [20]. The fraction of transmission that occurs
before symptom onset in symptomatically-infected individuals is defined by the cumulative
infectivity profile (or generation time) up to the time of symptom onset. The infectivity profile
and incubation periods are undefined (and unnecessary) for asymptomatic cases, and in the
model we make the simplifying assumption that the generation time distribution is the same
between asymptomatic and symptomatic cases.

Using the branching process model we calculate the number of infected individuals in the
second generation (secondary infections) and in the third generation (tertiary infections) after
the introduction of the first infected individual. We also keep track of the time at which the
transmission events occur. In our analysis we do not simulate the branching process explicitly,
but instead use a statistical description of the dynamics.

The expected number of secondary infections per infected depends on the transmissibility
of the virus (e.g. as captured by the basic reproductive number $R_0$), as well as the current level
of interventions in place to mitigate the spread of the virus. As we are interested in quantifying
the effects of TTIQ strategies, we introduce the parameter $R$ which represents the effective
reproductive number of the virus in the presence of interventions such as mask-wearing, social
distancing, school closures etc., but in the absence of isolation and quarantine. We refer to this
$R$ parameter as "*the baseline R-value in the absence of TTIQ*", and we have $R \leq R_0$ due to the

presence of the non-TTIQ preventative measures. Furthermore, the baseline reproductive number $R$ should be greater than or equal to the currently observed effective reproductive number, which includes the impact of in-place TTIQ measures. In the absence of TTIQ interventions, we would expect $R$ infections in the second generation, and $R^2$ infections in the third generation. This $R$-value will depend on the strain of SARS-CoV-2 that is considered, for example new variants of concern such as Alpha (B.1.1.7) and Delta (B.1.617.2) are significantly more transmissible than pre-existing variants [21–23]. The $R$-value will also be proportional to the size of the susceptible pool—which can be depleted due to death or acquired immunity—such that epidemic spread and vaccination campaigns will result in a smaller baseline $R$-value.

Furthermore, this $R$-value represents the average effective reproductive number across asymptomatic and symptomatic infections, i.e. $R = aR_a + (1 − a)R_s$, where $R_a$ and $R_s$ are the expected number of individuals who are directly infected by an asymptomatic or symptomatic individual in the absence of TTIQ, respectively. From the literature, we would expect that $R_a \leq R_s$ [7]. We can further define the parameter $\alpha = aR_a/R$ as the fraction of all transmission that originates from asymptomatically-infected individuals in the absence of TTIQ. This fraction has the property that if asymptomatic and symptomatic individuals are equally transmissive ($R_a = R_s$), then $\alpha$ is just the fraction of asymptomatic individuals ($\alpha = a$). If asymptomatic individuals are less infectious ($R_a < R_s$), then $\alpha < a$. Therefore the fraction of transmission from asymptomatic individuals in the absence of TTIQ satisfies $0 \leq \alpha \leq a$.

## Testing & isolating

Individuals who develop symptoms that are indicative of COVID-19 can be tested and subsequently isolated from the population. Testing & isolating acts to reduce the number of secondary infections per index case by shortening the duration in which the index case can infect susceptible individuals, i.e. isolation truncates the distribution of infection times (Fig 1A). We assume that isolation happens after a fixed delay of $\Delta_1$ days after symptom onset, and that only a fraction $f$ of symptomatic individuals are isolated. This incomplete coverage can be attributed to symptom misdiagnosis, a failure or unwillingness to get tested if symptomatic, a false-negative test result, or non-adherence to the isolation protocol. The fraction isolated $f$ can also be reduced by false-negative results based on potentially less-sensitive self-administered tests, which could prevent infected individuals from seeking confirmatory point-of-care tests. For those individuals who are isolated, we assume that they cannot infect further for the remaining duration of their infectious period. This assumption of perfect adherence to isolation once tested positive will lead to an overestimation of TTIQ effectiveness. Any lack of adherence to isolation could be accounted for in the model by reducing $f$, as long as this lack of adherence also means that their contacts are not traced. In the model, asymptomatic individuals are not subject to symptomatic testing and isolation. These asymptomatic individuals could be tested during surveillance (mass-testing or surge testing), and isolation would follow as above if a positive test result is obtained. However, in the scenario we are considering, only symptomatically-infected individuals will be tested and isolated.

The infected individuals who are not isolated will infect $R_a$ or $R_s$ secondary contacts, depending on whether they are classified as asymptomatic or symptomatic. Isolated symptomatic cases will infect $P(\Delta_1) \times R_s$ secondary contacts, where $0 \leq P(\Delta_1) \leq 1$ is the cumulative fraction of the infection time distribution that lies before the isolation time $\Delta_1$ (see Fig 1A). Averaging across these scenarios gives the expected number of secondary infections per infected case. We can repeat this analysis for each of the secondary infections to calculate the number tertiary infections under testing & isolation. See S1 Appendix for the full calculation.

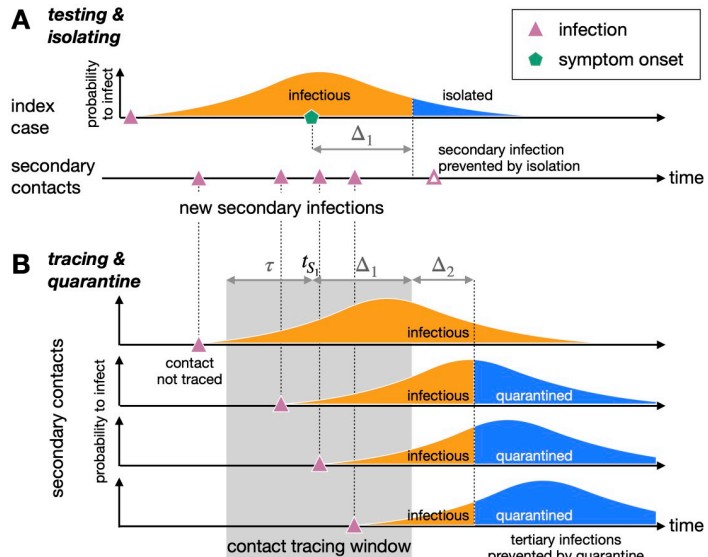

**Fig 1. Quantifying the impact of TTIQ interventions using a mathematical model.** A: Under testing & isolation, index cases are identified and isolated from the population after a delay $\Delta_1$ after they develop symptoms (at time $t_{S_1}$). This curtails their duration of infectiousness and reduces the number of secondary infections. This isolation occurs in a fraction $f$ of symptomatic individuals. B: Under additional contact tracing & quarantine, the contacts of an index case can be identified and quarantined after an additional delay $\Delta_2$. This reduces the onward transmission from these secondary contacts. Only contacts that occur during the contact tracing window can be identified. This window extends from $\tau$ days before the index case developed symptoms (i.e. $t_{S_1} - \tau$) to the time at which the index case was isolated (i.e. $t_{S_1} + \Delta_1$). A fraction $g$ of the contacts who were infected within the contact tracing window are quarantined. The remaining individuals are not quarantined, but could be isolated if they are later detected as an index case. The distributions shown here are schematic representations of the infectivity profile and/or generation time interval, which are quantitatively displayed in Fig I in S1 Appendix. These distributions reflect an individual's infectiousness as a function of time.

## Contact tracing & quarantine

When a symptomatic index case is identified by testing, they can be interviewed by contact tracers to determine whom they have potentially infected, with the aim of quarantining these exposed individuals. The contact tracers focus on a specific time window of infection to identify the contacts with highest risk of being infected. In our model this window extends to $\tau$ days before symptom onset in the index case. It is unlikely that every contact within this time window is memorable or traceable, so we assume that only a fraction $g$ of the secondary contacts within this window are eventually quarantined. Quarantine begins after a fixed delay of $\Delta_2$ days after the isolation of the index case, representing the time required for the contact to be identified by contact tracers and to enter quarantine. For those who are quarantined, we assume that they cannot infect further for the remaining duration of their infectious period. This assumption of perfect adherence to quarantined once identified through contact tracing will lead to an overestimation of TTIQ effectiveness. However, any lack of adherence to isolation is easily accounted for in the model by reducing $g$. Importantly, quarantine occurs independently of whether these secondary infections are asymptomatic or will eventually develop symptoms. Furthermore, contact tracing can also achieved through digital app-based technology [12]. The proposed model applies to both manual and digital contact tracing, but we note that we would expect different parameter combinations for digital versus manual contact tracing, for example reduced delays $\Delta_2$ for digital contact tracing [12, 13].

Quarantine shortens the duration in which the identified secondary contacts can transmit further to tertiary contacts (Fig 1B). For each quarantined secondary contact, we calculate the number of onward infections by computing the cumulative fraction of the infection time distribution before quarantine begins and multiplying by $R_a$ or $R_s$, depending on whether the secondary contact is classified as asymptomatic or symptomatic.

If a secondary contact is not quarantined, then they could be detected after symptom onset (if not asymptomatic) as a new index case and subsequently isolated. The number of onward infections that result from these individuals is typically higher than for those who are quarantined, as they have to wait until symptom onset before they can be isolated, or they may not develop symptoms at all in which case they are not isolated. Finally, some secondary contacts are not quarantined or isolated, and will infect $R_a$ or $R_s$ tertiary contacts. The average number of infections caused by the quarantined, isolated, and non-isolated secondary contacts is then the number of tertiary infections per index case. See S1 Appendix for the full calculation.

## TTIQ parameters

The efficacy of the TTIQ interventions depends on how quickly and accurately they are implemented. To this end, we have introduced five parameters to describe the TTIQ process (Table 1). We systematically explore this TTIQ parameter space, first for the testing & isolation intervention in the absence of contact tracing (Fig 1A), and then with additional tracing & quarantine (Fig 1B).

Due to the high between-country variability of testing coverage (*f*), contact tracing success (*g*), and the respective delays, as well as the lack of publicly-available data on these topics, we keep these values as free parameters in our analyses.

In all analyses we focus on fixed TTIQ parameter values for all individuals in the branching process, as opposed to sampling each individual's parameters from a distribution. This simplifies the visualisation and interpretation of results. We expect that the averaged results when using distributed parameters would closely reflect our fixed-value results, but would lead to increased variance/uncertainty in our estimates. Heterogeneity in the individuals' baseline reproductive number (due to contact number and transmission heterogeneities) is addressed in S3 Appendix.

## Effective reproduction number

The effectiveness of the TTIQ intervention can be quantified by calculating the effective reproduction number in the presence of the interventions, $R_{\mathrm{TTIQ}}$, which describes the expected number of secondary infections per infected individual. If $R_{\mathrm{TTIQ}} > 1$ then the epidemic is

**Table 1. Parameter definitions for the TTIQ interventions.**

| Parameter | Description | Range |
|---|---|---|
| *f* | Probability that a symptomatic individual is isolated from the population | $0\% \leq f \leq 100\%$ |
| $\Delta_1$ | Time delay between symptom onset and isolation | $\Delta_1 \geq 0$ days |
| $\tau$ | Duration prior to symptom onset in which contacts are identifiable | $\tau \geq 0$ days |
| *g* | Fraction of identifiable contacts that are successfully traced and quarantined per isolated index case | $0\% \leq g \leq 100\%$ |
| $\Delta_2$ | Time delay between isolation of the index case and the start of quarantine for the secondary contacts | $\Delta_2 \geq 0$ days |

The delay and lookback parameters $\Delta_1$, $\Delta_2$, and $\tau$ are illustrated in Fig 1.

growing, while a value of less than one means the epidemic is being suppressed. For our branching process model, we define the reproductive number as $R_{\text{TTIQ}} = n_3/n_2$, where $n_2$ and $n_3$ are the expected number of secondary and tertiary infections per index case, respectively. In other words, we define the reproductive number as the average number of infecteds in the third generation per infected in the second generation. It is necessary to work with the third generation (as opposed to just the first and second generations) as this is where the impact of contact tracing and quarantine is first observed. Under strategies of testing & isolation alone (i.e. no contact tracing), we use the notation $R_{\text{TI}}$ for clarity.

## Computing uncertainties

As shown in Fig I in S1 Appendix, there is considerable uncertainty in the variance of the inferred generation time distribution and infectivity profile. We propagate this uncertainty into our calculation of $R_{\text{TTIQ}}$. Briefly, we sample parameter combinations that make up the 95% confidence interval of the generation time distribution and infectivity profile, and then compute $R_{\text{TTIQ}}$ for each parameter set. The maximum and minimum of these values then describe the confidence interval for the level of transmission. Complete details are provided in S1 Appendix.

## Interactive app

To complement the results in this manuscript, and to allow readers to investigate different TTIQ parameter settings, we have developed an online interactive application. This can be found at https://ibz-shiny.ethz.ch/covidDashboard/ttiq.

## Results

### Reducing transmission by testing & isolating symptomatic cases

Based on our transmission model, testing & isolating of symptomatic cases alone (i.e. without additional contact tracing & quarantine) is capable of suppressing epidemic growth ($R_{\text{TI}} < 1$) with a *baseline R-value in the absence of TTIQ* of up to 1.76 [95% confidence interval (CI): 1.57,1.98], assuming that asymptomatic individuals contribute $\alpha = 20\%$ of infections (Fig II in S2 Appendix). To achieve this level of suppression, each symptomatic individual ($f = 100\%$) would have to isolate immediately at symptom onset ($\Delta_1 = 0$ days), representing the upper limit of testing & isolation performance. We again note that the baseline $R$ parameter depends on the current suppression measures against SARS-CoV-2 transmission (social distancing, mask wearing, home office, etc., but not TTIQ interventions), as well as seasonality, the variant under consideration, and levels of immunity/vaccination. Importantly, this predicted upper limit of $R = 1.76$ is below the estimated $R_0$ of SARS-CoV-2 ($R_0 \approx 2.5$ [9]—5.7 [24]). Hence, testing & isolating of symptomatic cases alone as a control strategy would not have been sufficient to prevent epidemic growth, even before the emergence of more transmissible variants. One would have to reduce the number of susceptible individuals in the population by at least 30% (e.g. through vaccination) for testing & isolating alone to be a viable strategy.

The region of ($f$, $\Delta_1$) parameter space in which $R_{\text{TI}}$ is less than one, i.e. the region in which an epidemic can be controlled by testing & isolating, is shrinking for higher $R$ epidemics (Fig 2A). Higher testing & isolation coverage ($f$) or shortened delays between symptom onset and isolation ($\Delta_1$) are required to control SARS-CoV-2 outbreaks as $R$ increases. By increasing the fraction of symptomatic individuals that are isolated ($f$), there can be a greater delay to isolation without any increase in $R_{\text{TI}}$, but with diminishing returns.

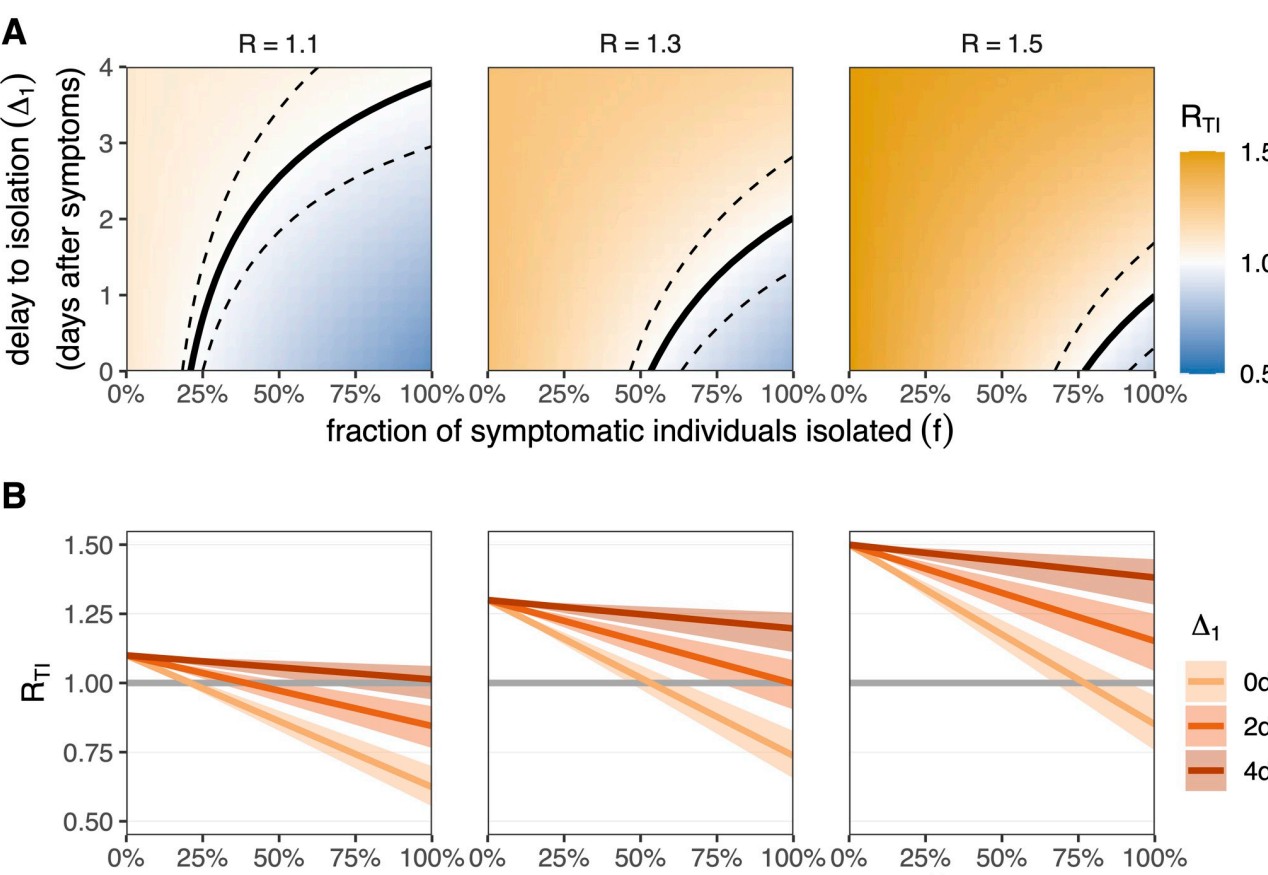

**Fig 2. The reproductive number $R_{TI}$ under testing & isolation only.** A: The impact of testing & isolation on $R_{TI}$ as a function of the fraction of symptomatic individuals that are isolated ($f$; x-axis) and delay to isolation after symptom onset ($\Delta_1$; y-axis) for different baseline $R$ values (columns). The black line represents the critical reproductive number $R_{TI} = 1$. Above this line (orange zone) we have on average more than one secondary infection per infected and the epidemic is growing. Below this line (blue zone) we have less than one secondary infection per infected and the epidemic is suppressed. Dashed lines are the 95% confidence interval for this threshold, representing the uncertainty in the inferred generation time distribution and infectivity profile. B: Lines correspond to slices of panel A at a fixed delay to isolation $\Delta_1 = 0$, 2, or 4 days after symptom onset (colour). Shaded regions are 95% confidence intervals for the reproductive number, representing the uncertainty in the inferred generation time distribution and infectivity profile. Horizontal grey line is the threshold for epidemic control ($R_{TI} = 1$). We fix the fraction of transmission that is attributed to asymptomatic infections to $\alpha = 20\%$, where asymptomatic individuals are not tested or isolated. Data provided in S1 Dataset.

A SARS-CoV-2 outbreak with $R = 1.1$ (in the absence of TTIQ) can be controlled by isolating as few as 21% [95% confidence interval (CI): 18%,25%] of symptomatic individuals at the time of symptom onset ($\Delta_1 = 0$ days) (Fig 2B). If the symptomatic individuals wait $\Delta_1 = 2$ days after symptom onset before isolating (i.e. they wait for a test result), then 39% [CI: 30%,54%] of symptomatic infecteds would have to be isolated for the epidemic to be controlled. Isolating after $\Delta_1 = 4$ days would be insufficient to control the epidemic even if all symptomatic individuals were isolated [CI: 63%,n.a.]. For faster-spreading SARS-CoV-2 outbreaks or more transmissible variants (baseline $R = 1.5$ in the absence of TTIQ), we would require 77% [CI: 67%,91%] of symptomatic infecteds to be isolated immediately after they develop symptoms ($\Delta_1 = 0$ days) to control the epidemic. With a delay $\Delta_1 \geq 2$ days, testing & isolating of symptomatic cases would be insufficient to control the epidemic, even if 100% of symptomatic infecteds are isolated.

We have predominantly focussed on $\alpha = 20\%$ of infections being attributable to persistently-asymptomatic individuals in the absence of TTIQ. As this fraction increases, we observe a linear increase in $R_{TI}$, i.e. increasing the fraction of transmission that is attributable to asymptomatic infections leads to reduced efficacy of testing & isolating, as fewer cases are identified by testing only symptomatics (Fig IIIA in S2 Appendix). It should be noted that under testing & isolation measures, a larger fraction of onward transmission is attributable to asymptomatic infections when compared to the scenario of no TTIQ (Fig IIIB in S2 Appendix).

## Reducing transmission by additional contact tracing & quarantine

With additional contact tracing & quarantine, the theoretical upper limit of TTIQ efficacy is greatly increased compared to testing & isolation alone: under perfect conditions with all contacts quarantined immediately after symptom onset in the index case, TTIQ can suppress epidemics with a *baseline R-value in the absence of TTIQ* of up to 4.24 [95% CI: 3.10,5.83] (Fig II in S2 Appendix for $\alpha = 20\%$). However, it is unlikely that such a high level of suppression could be achieved in practice due to delays and inaccuracies in the contact tracing process.

Under ideal TTIQ conditions, additional tracing & quarantine can more than double the effectiveness of the intervention compared to testing & isolation alone (Fig II in S2 Appendix). However, under more realistic expectations of inaccuracies and delays in the TTIQ processes, the majority of transmission is prevented by testing & isolation if less than $g = 60\%$ of contacts are quarantined (S1 Fig).

We can visualise the additional benefit that contact tracing & quarantine brings to testing & isolation in our model by gradually increasing the fraction of contacts that are quarantined, $g$. For $g = 0$, no contacts are traced & quarantined, and hence we return to the testing & isolation strategy (Fig 2). By increasing $g$, we expand the parameter space in which $R_{TTIQ} < 1$ (Fig 3),

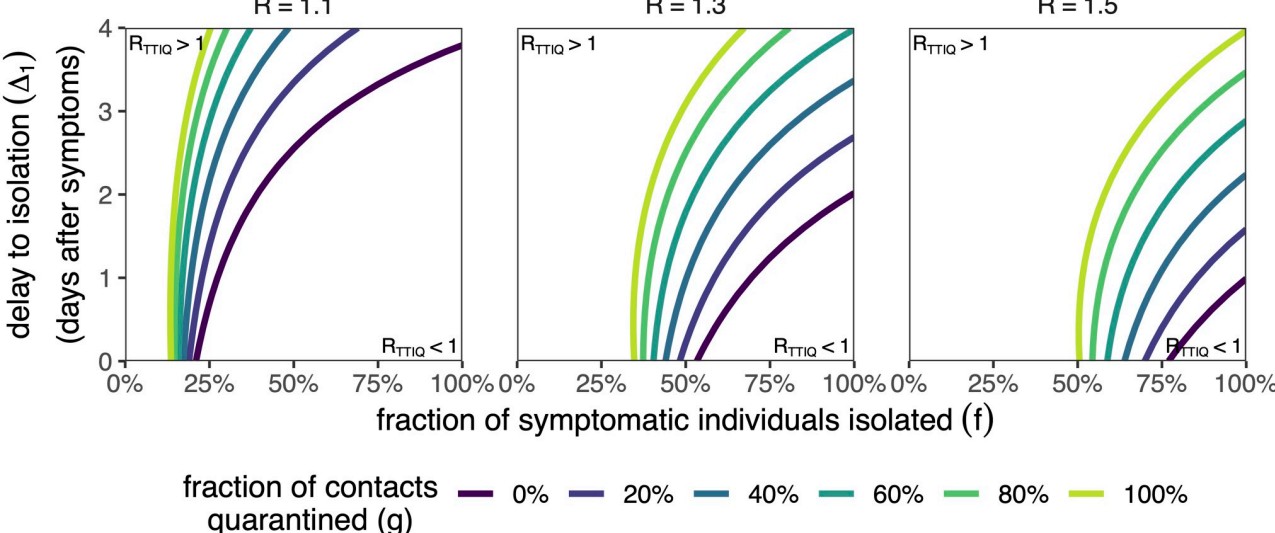

**Fig 3. The impact of tracing & quarantine on the reproductive number.** The impact of tracing & quarantine on the reproductive number $R_{TTIQ}$ as a function of the fraction of symptomatic individuals that are isolated ($f$; x-axis) and delay to isolation after symptom onset ($\Delta_1$; y-axis), for different contact tracing & quarantine success probabilities $g$ (colour) across different baseline $R$ values (columns). We fix $\Delta_2 = 2$ days and $\tau = 2$ days. The contours separate the regions where the epidemic is growing ($R_{TTIQ} > 1$; top-left) and the epidemic is suppressed ($R_{TTIQ} < 1$; bottom-right). The contours for $g = 0$ are equivalent to the contours in Fig 2. We fix the fraction of transmission that is attributed to asymptomatic infections to $\alpha = 20\%$. Asymptomatic individuals are not tested or isolated, but are subject to quarantine after contact tracing. We do not show confidence intervals for clarity of presentation. Data provided in S1 Dataset.

i.e. contact tracing allows an epidemic to be controlled for lower fractions of index cases found ($f$) and/or longer delays to isolating the index case after they develop symptoms ($\Delta_1$). Furthermore, for a given set of testing & isolation parameters $f$ and $\Delta_1$, we can control higher $R$-value epidemics with contact tracing & quarantine that would be otherwise uncontrollable.

To obtain a systematic understanding of the impact that each parameter of the TTIQ process has on the effective reproductive number $R_{\mathrm{TTIQ}}$, we can individually vary each of the five TTIQ parameters. To this end, we calculate $R_{\mathrm{TTIQ}}$ for focal parameter sets of ($f, g, \Delta_1, \Delta_2, \tau$). We then perturb each single parameter, keeping the remaining four parameters fixed, and compute the new value of $R_{\mathrm{TTIQ}}$ (Fig 4).

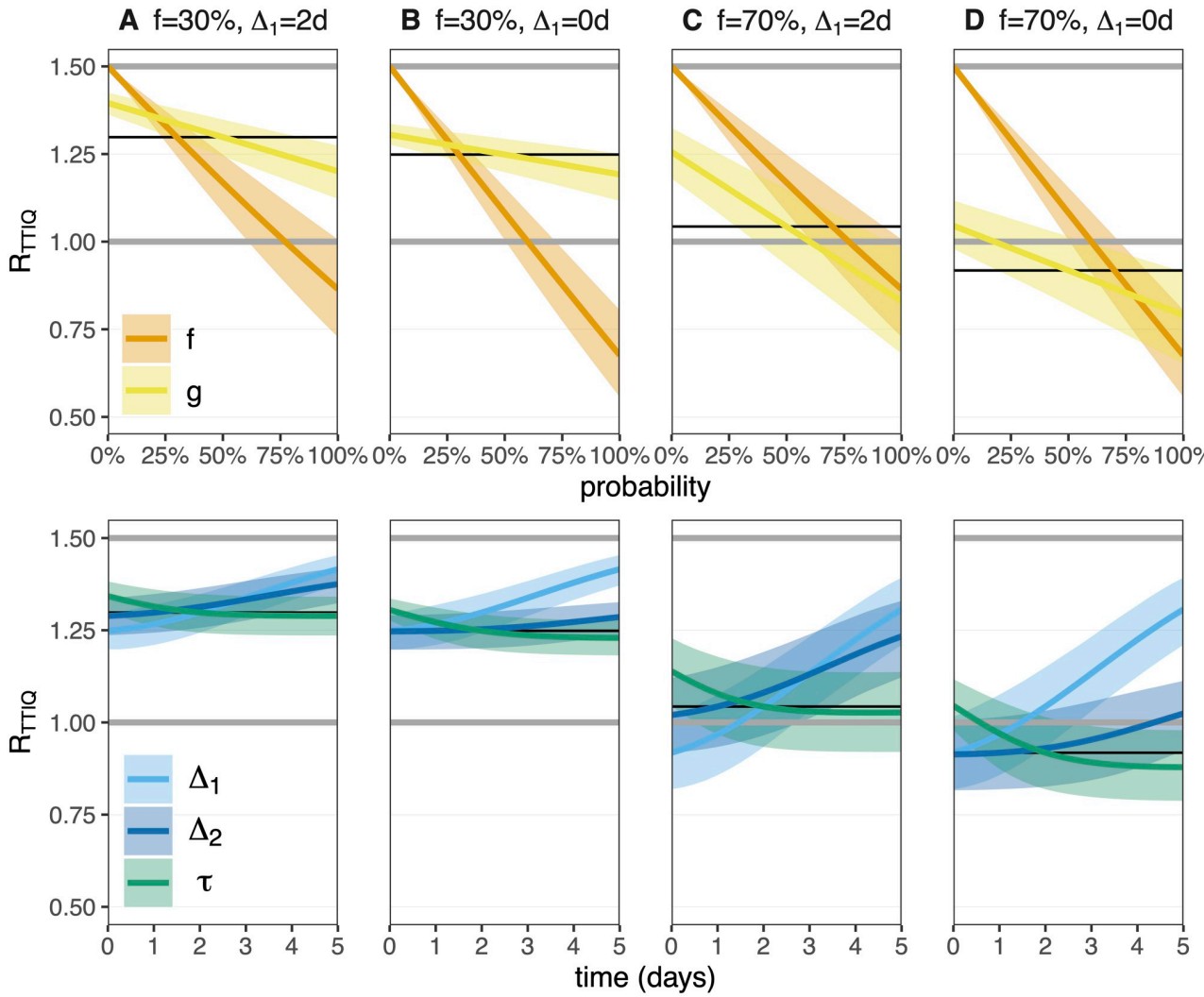

**Fig 4. The response of the reproductive number $R_{\mathrm{TTIQ}}$ to single TTIQ parameter perturbations.** We set the baseline $R = 1.5$ throughout, which is the intensity of the epidemic in the absence of any TTIQ intervention. We consider four focal TTIQ parameter combinations, with $f \in \{30\%, 70\%\}$, $\Delta_1 \in \{0, 2\}$ days, $g = 50\%$, $\Delta_2 = 1$ day, and $\tau = 2$ days. $R_{\mathrm{TTIQ}}$ for the focal parameter sets are shown as thin black lines. With $f = 0$ (no TTIQ) we expect $R_{\mathrm{TTIQ}} = R$ (upper grey line). We then vary each TTIQ parameter individually, keeping the remaining four parameters fixed at the focal values. The upper panel shows the probability parameters $f$ and $g$, while the lower panel shows the parameters which carry units of time (days). The critical threshold for controlling an epidemic is $R_{\mathrm{TTIQ}} = 1$ (lower grey line). We fix the fraction of transmission that is attributed to asymptomatic infections to $\alpha = 20\%$. A sensitivity analysis to $\alpha$ is shown in Fig I in S2 Appendix. Asymptomatic individuals are not tested or isolated, but are subject to quarantine after contact tracing. Data provided in S1 Dataset.

Modifying the fraction of symptomatic index cases that are identified and isolated ($f$) has the largest effect of all parameter changes. By identifying more index cases (increasing $f$), we not only prevent the onward transmission to secondary contacts through isolation, but we also allow infected contacts to be traced and quarantined.

Increasing the fraction of secondary contacts that are quarantined ($g$) has a smaller benefit than increasing $f$. If only 30% of symptomatic index cases are identified, then increasing $g$ results in a small reduction of $R_{\mathrm{TTIQ}}$ and for $R = 1.5$ the epidemic cannot be controlled even if all secondary contacts ($g = 100\%$) of known index cases are quarantined (Fig 4A & 4B). However, if a large fraction of symptomatic index cases are identified ($f = 70\%$), then increasing $g$ can control an epidemic that would be out of control in the absence of contact tracing (Fig 4C & 4D).

After increasing $f$, the next most effective control strategy is to reduce the delay between symptom onset and isolation of the index case ($\Delta_1$). Reducing the time taken to quarantine secondary contacts ($\Delta_2$) has a lesser effect on $R_{\mathrm{TTIQ}}$. Finally, looking back further while contact tracing (increasing $\tau$) allows more secondary contacts to be traced and quarantined. However, this does not translate into a substantial reduction in $R_{\mathrm{TTIQ}}$ as the extra contacts which are traced have already been infectious for a long time, and will thus have less remaining infectivity potential to be prevented by quarantine. Hence increasing $\tau$ comes with diminishing returns.

To check the robustness of these effects across all parameter combinations (not just perturbing a single parameter), we randomly sampled parameter combinations ($f$, $g$, $\Delta_1$, $\Delta_2$, $\tau$) and used linear discriminant analysis (LDA) to capture the impact that each parameter has on $R_{\mathrm{TTIQ}}$ (Fig 5). We find that $f$ is the dominant parameter to determine the reproductive number, followed by $\Delta_1$, $g$, $\Delta_2$, and finally $\tau$ has the smallest impact. Furthermore, by looking at the distribution of the randomly-sampled TTIQ parameters across different $R_{\mathrm{TTIQ}}$ values (S2 Fig), we observe that low $f$ values are strongly associated with low TTIQ effectiveness (although a high $f$ value is not necessarily associated with high effectiveness).

The output of the LDA analysis is dependent on the range of parameter values from which we sample (S3 Fig). While $f$ and $g$ are naturally bounded from 0% to 100%, the time-valued parameters $\Delta_1$, $\Delta_2$, and $\tau$ have no natural upper limit. Without empirical data to inform these prior distributions, we focus on durations from zero to five days as shown in Fig 4. Furthermore, as a linear approximation the LDA does not capture the effect of covariance between parameters. To capture these parameter interactions, we can also include quadratic terms (e.g. $f \times g$) as independent parameters in the LDA. From this analysis (S4 Fig), we see that the terms $f \times \Delta_1$ and $g \times \Delta_2$ correlate positively with $R_{\mathrm{TTIQ}}$, such that increasing the delays $\Delta_1$ and $\Delta_2$ can negate the increase in TTIQ efficacy that is bought by increasing $f$ or $g$, respectively.

Finally, we comment on the role of asymptomatic transmission across the TTIQ intervention. Although quarantine of a traced contact occurs independently of whether that contact will be symptomatic or asymptomatic, the probability that the contact is identified in the first place depends on whether the infector is asymptomatic or not. Hence, TTIQ will decrease in effectiveness as the fraction of transmission that is attributable to asymptomatic individuals ($\alpha$) increases (Fig IIA in S2 Appendix).

## Discussion

By combining empirically well-supported estimates of the infection timing of SARS-CoV-2 with a simple model of transmission dynamics, we have calculated the impact of test-trace-isolate-quarantine (TTIQ) interventions against the spread of COVID-19. Under idealised conditions, testing & isolation plus contact tracing & quarantine can prevent substantially more transmission than testing & isolation only. However, the effects of delays and inaccuracies in the TTIQ processes are compounded for contact tracing & quarantine, which ultimately relies

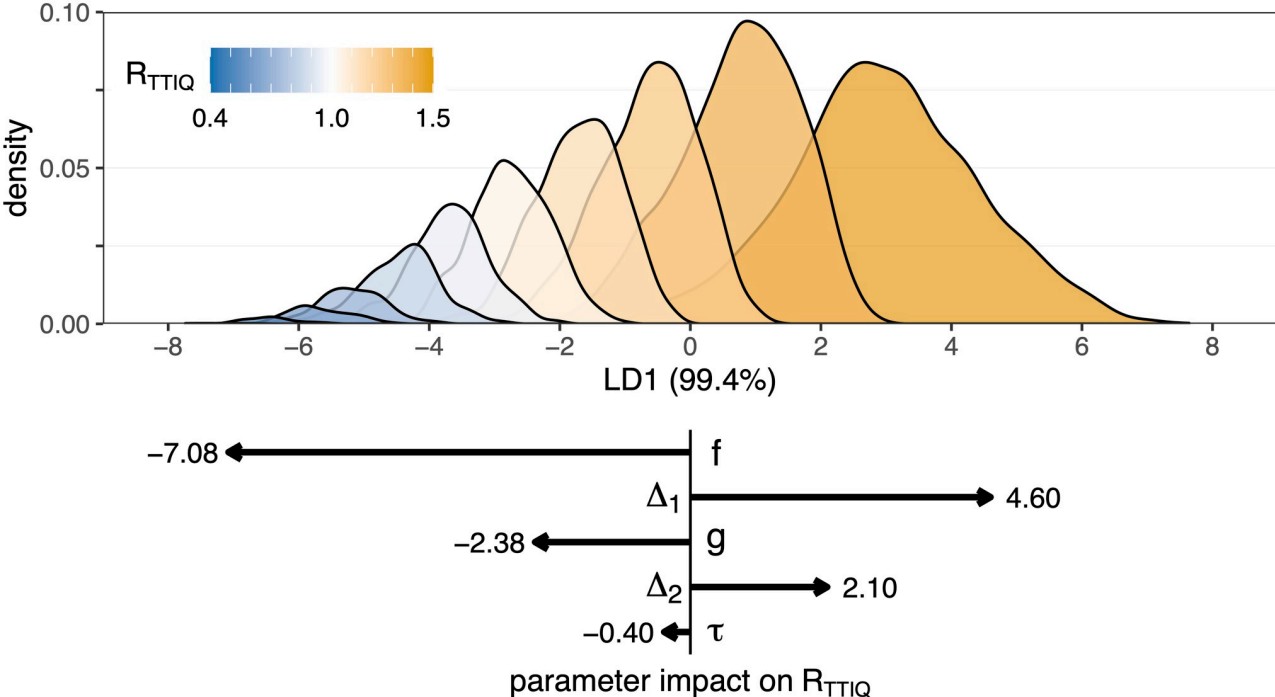

**Fig 5. Linear discriminant analysis (LDA) of the impact of TTIQ strategies on the reproductive number $R_{\text{TTIQ}}$.** We fix the baseline $R = 1.5$ and $\alpha = 20\%$, and then we randomly uniformly sample 10,000 parameter combinations from $f \in [0\%, 100\%]$, $g \in [0\%, 100\%]$, $\Delta_1 \in [0, 5]$ days, $\Delta_2 \in [0, 5]$ days, and $\tau \in [0, 5]$ days. The reproductive number is calculated for each TTIQ parameter combination, and the output ($R_{\text{TTIQ}}$) is categorised into bins of width 0.1 (colour). We then use LDA to construct a linear combination (LD1) of the five (normalised) TTIQ parameters which maximally separates the output categories. We then predict the LD1 values for each parameter combination, and construct a histogram of these values for each category. The lower panel shows the components of the primary linear discriminant vector (LD1). By multiplying the (normalised) TTIQ parameters by the corresponding vector component, we arrive at the LD1 prediction which corresponds to the predicted reproductive number under that TTIQ strategy. Longer arrows (larger magnitude components) correspond to a parameter having a larger effect on the reproductive number. The distributions of parameters per categorised reproductive number is shown in S2 Fig. Data provided in S1 Dataset.

on index case identification to be effective. Ignoring this compounding effect would potentially overestimate the impact that contact tracing can have on transmission reduction. Based on our systematic analysis, we find that the greatest improvement to the TTIQ process would come from increased identification and isolation of symptomatic index cases and reduction of delay between symptom onset and isolation. These parameters contribute to the direct reduction of onward infection from an index case, and optimising them allows more contacts to be traced earlier. These results align with those of Grantz et al. [17]. From a public health perspective, increasing the identification and speeding up the isolation of symptomatic index cases could be achieved through widely-available rapid testing. Despite the potentially lower sensitivity of rapid tests compared to RT-PCR tests, their effectiveness at reducing transmission has been demonstrated in simulation studies of index case isolation [25].

Increasing the duration of the contact tracing window by looking back further in time has limited return under our model of forward contact tracing (identifying who was infected by the index case). However, if we were interested in identifying the source of infection (backwards contact tracing), then increasing the duration of the contact tracing window could lead to the identification of transmission clusters.

When comparing to the findings of Ferretti et al. [12], we find that contact tracing has less impact on epidemic suppression, and that the speed of contact tracing is of secondary

importance to the speed of isolating index cases. This difference can be attributed to Ferretti et al.'s [12] use of Fraser et al.'s [26] approach to model contact tracing and isolation as independent events (i.e. tracing an index cases' contacts says nothing about whether the index case has been isolated). Although this assumption leads to analytically tractable predictions of the reproductive number under TTIQ, it also leads to an overestimation of contact tracing's impact [26]. Our approach can therefore be considered as a methodological advance over Fraser et al. [26] and should be employed in the analysis of future epidemic scenarios.

Kretzschmar et al. [13]—this time with contact tracing dependent on testing & isolation—concluded that reducing the delay to isolation after symptom onset has the greatest impact on TTIQ effectiveness. They further showed that the effective reproductive number was insensitive to varying the testing coverage, although only at a fixed delay of four days between symptom onset and index case isolation. Based on our systematic LDA analysis with quadratic parameters (S4 Fig), we know that there is considerable interaction between testing coverage $f$ and isolation delay $\Delta_1$. Therefore, we expect that sensitivity to testing coverage would appear at shorter delay values, and on average across these parameters we show that increasing $f$ has a greater effect on the reproductive number than decreasing $\Delta_1$.

Our approach and results are crucially dependent on the distribution of infection times (generation time and infectivity profile) and although we have used well-supported estimates, there are inherent limitations to deriving these distributions based on transmission pairs. These transmission pairs are representative of symptomatic cases, but the infectiousness profiles for persistently-asymptomatic infections are as-yet unknown [6]. We have assumed that asymptomatically-infected individuals have the same infection timing distributions as symptomatic individuals, but any differences between the shapes of these profiles will lead to different results in terms of transmission reduction. Uncertainty in the inferred infection timing distributions is carried through our analysis and is captured by the confidence intervals shown in the figures and reported in the text. Furthermore, we do account for potential differences in the overall transmissibility between asymptomatic and symptomatic individuals. It is possible that the 20% of infections that are asymptomatic are responsible for less than 20% of transmission in the absence of any TTIQ interventions. We show in S2 Appendix that TTIQ becomes more effective as asymptomatic transmission decreases. Therefore, our results could be underestimating TTIQ efficacy, to a small extent.

Our model is parametrised on distributions of the timing of transmission estimated prior to the emergence of new, more transmissible variants. If new variants simply have higher transmissibility—without changes in the timing of transmission—our fundamental analysis remains the same. In this case, TTIQ may be insufficient to control the spread of highly-transmissible (higher $R$-value) new variants, as captured in Figs 2 and 3. If the increased transmission of the new variants is due to a longer-lasting infectious period, then we expect TTIQ to be more efficient, as the additional transmission events would be prevented by isolation and quarantine. If the new variants are more transmissible during early (presymptomatic) infection, then we expect the relative benefit of contact tracing over testing & isolating to increase.

In terms of modelling the TTIQ process, we have assumed that identified symptomatic index cases are isolated after symptom onset and have their contacts traced. If the index case fails to adhere to the isolation protocol, then we will overestimate the amount of transmission prevented by isolation. However, uncertainty in whether contacts adhere to quarantine protocols, or whether contact tracers actually identify contacts, is captured in the parameter $g$. Lower adherence to quarantine or missed contacts due to overwhelmed contact tracers is captured by lowering $g$. Hill et al. [27] recently demonstrated in a network modelling study that increased adherence to isolation and contact tracing measures can significantly reduce the size of an outbreak.

In our approach, we assume a baseline $R$ that is defined in the absence of the modelled TTIQ intervention (i.e. no testing, isolation, or contact tracing). The empirical value for this baseline $R$ is not known, as observed values of the reproductive number in most countries include the impact of the modelled intervention. In itself, this does not impact the result of our analysis: the impact of isolation remains higher than that of quarantine across different values of $R$ (Fig 3). However, in contexts where a large proportion of symptomatic individuals already isolate, the scope for increasing isolation may be limited. Under such circumstances, mass or random testing, if successfully followed by isolation, may be a promising intervention. This is supported by data on the effectiveness of mass testing interventions, for example in Slovakia [28]. In this scenario it would be possible to identify asymptomatic index cases, as well as identifying eventually-symptomatic cases before symptom onset. Through this increased index case identification and isolation, as well as the reduced time that these index cases are infectious and non-isolated, and also reducing the number of secondary contacts that have to be identified by contact tracing, mass/random testing would therefore increase the overall performance of TTIQ.

Our analysis is, to some extent, limited by the assumptions which underlie the branching process framework. The infinite population size assumption prevents us from computing the fraction of population that is infected, or from a socioeconomic point of view the fraction of population isolated/quarantined at a given time. Furthermore, with the branching process we cannot observe long-term effects caused by depleting susceptibles through quarantine, immunity, or death. However, the branching process approach is valid over short time scales (like the two generations of transmission that we calculate), provided that the susceptible population size is much larger than $R^2$. The effect of susceptible depletion can also be incorporated by lowering the baseline reproductive value $R$ in the model.

Assuming a fixed value of the baseline reproductive value $R$ is a further limitation of our approach, as the impact of overdispersion of contact number distributions and superspreading is well documented for infectious disease dynamics [29]. If we were to sample $R$ for the index case and each secondary case from identical overdispersed negative binomial distributions, then the expectation value would be unchanged from our current approach: only the variance/ uncertainty in our predictions would increase (S3 Appendix). The equivalence of expectation values could break down if we were to assume a finite capacity of contact tracing, such that the quarantined fraction of contacts of index cases with a large individual reproductive number may be less than $g$.

## Conclusion

Here we have shown through systematic analysis how the TTIQ processes can be optimised to bring the effective reproductive number below one. Crucially, contact tracing & quarantine adds security to testing & isolating strategies, where high coverage and short delays are necessary to control an epidemic. By improving the testing & isolation coverage and reducing the delay to index case isolation, we can greatly increase the efficacy of the overall TTIQ strategy.

## Supporting information

**S1 Appendix. Supplementary materials and methods.** This file contains mathematical details of the model and analysis.
(PDF)

**S2 Appendix. Supplementary results I.** Impact of asymptomatics.
(PDF)

**S3 Appendix. Supplementary results II.** Overdispersion.
(PDF)

**S1 Fig. Quarantine versus isolation.** The fraction of prevented transmission that can be attributed to quarantine, rather than isolation. Let $R_{\text{TTIQ}}(g)$ be the reproductive number in the presence of TTIQ interventions in which a fraction $g$ of contacts of identified index cases are quarantined. In the absence of TTIQ measures, we expect a reproductive number of $R$. We then define $Y(g) = R - R_{\text{TTIQ}}(g)$ as reduction of transmission due to TTIQ, and $Y(0) = R - R_{\text{TTIQ}}(0)$ as the reduction of transmission due only to isolation (i.e. no contact tracing & quarantine). We then define the fraction of prevented transmission due to quarantine as $[Y(g) - Y(0)]/Y(g)$, which we plot as a function of $g$. Note that we are computing how much extra transmission is prevented by quarantine, which may just be one days worth of transmission before the contact becomes symptomatic and would anyway be isolated. We vary the fraction of symptomatic index cases that are isolated $f$ (colour), and we fix $\Delta_1 = \Delta_2 = \tau = 2$ days. We further fix the fraction of transmission that is attributed to asymptomatic infections to $\alpha = 20\%$ and $R = 1.5$ (although the fraction shown is independent of $R$). Above the horizontal line, more transmission is prevented by quarantine than by isolation. Data provided in S1 Dataset.
(PDF)

**S2 Fig. LDA parameter distributions.** The distributions of (normalised) parameters per categorised group of $R_{\text{TTIQ}}$ as used in the LDA analysis in Fig 5 in the manuscript. We uniformly sample 10,000 parameter combinations from $f \in [0\%, 100\%]$, $g \in [0\%, 100\%]$, $\Delta_1 \in [0, 5]$ days, $\Delta_2 \in [0, 5]$ days, and $\tau \in [0, 5]$ days. The reproductive number $R_{\text{TTIQ}}$ is calculated for each parameter combination and categorised into bins of width 0.1 (colour). The upper row shows how many parameter combinations resulted in each category of $R_{\text{TTIQ}}$. The next five rows show how the parameters are distributed within each category, while the horizontal bar shows the median parameter value. We fix $R = 1.5$ and $\alpha = 20\%$. Data provided in S1 Dataset.
(PDF)

**S3 Fig. LDA range sensitivity.** Impact of varying the range from which we sample time-dependent parameters on the LDA output (without quadratic terms). Each bar represents the magnitude of the components of the primary linear discriminant vector (LD1) for each parameter (colour). Data provided in S1 Dataset.
(PDF)

**S4 Fig. LDA with quadratic terms.** Linear discriminant analysis (LDA) of the impact of TTIQ strategies on the reproductive number $R_{\text{TTIQ}}$, now including quadratic terms. We use the same uniformly-sampled data as in Fig 5 in the manuscript, but now we include the quadratic parameter terms (e.g. $f \times g$) as discriminators too. We take the square root of these quadratic terms to ensure the parameter distributions are not overly skewed. We then use LDA to construct a linear combination (LD1) of the now 15 TTIQ parameters which maximally separates the output categories. We then predict the LD1 values for each parameter combination, and construct a histogram of these values for each category. The lower panel shows the components of the primary linear discriminant vector (LD1). By multiplying the (normalised) TTIQ parameters by the corresponding vector component, we arrive at the LD1 prediction which corresponds to the predicted reproductive number under that TTIQ strategy. Longer arrows (larger magnitude components) correspond to a parameter having a larger effect on the reproductive number. Data provided in S1 Dataset.
(PDF)

**S1 Dataset. Raw data for quantitative figures.**
(ZIP)

## Author Contributions

**Conceptualization:** Peter Ashcroft, Sonja Lehtinen, Sebastian Bonhoeffer.

**Data curation:** Peter Ashcroft.

**Formal analysis:** Peter Ashcroft.

**Funding acquisition:** Sebastian Bonhoeffer.

**Investigation:** Peter Ashcroft.

**Methodology:** Peter Ashcroft, Sonja Lehtinen, Sebastian Bonhoeffer.

**Project administration:** Peter Ashcroft, Sebastian Bonhoeffer.

**Software:** Peter Ashcroft.

**Supervision:** Sebastian Bonhoeffer.

**Validation:** Peter Ashcroft, Sonja Lehtinen, Sebastian Bonhoeffer.

**Visualization:** Peter Ashcroft.

**Writing – original draft:** Peter Ashcroft, Sonja Lehtinen, Sebastian Bonhoeffer.

**Writing – review & editing:** Peter Ashcroft, Sonja Lehtinen, Sebastian Bonhoeffer.

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
