## [Decision Letter · Decision Letter 0]

5 Oct 2021

PONE-D-21-26262Test-trace-isolate-quarantine (TTIQ) intervention strategies after symptomatic COVID-19 case identificationPLOS ONE

Dear Dr. Ashcroft,

Thank you for submitting your manuscript to PLOS ONE. After careful consideration, we feel that it has merit but does not fully meet PLOS ONE’s publication criteria as it currently stands. Therefore, we invite you to submit a revised version of the manuscript that addresses the points raised during the review process. Additional Editor Comments:

I found this manuscript to be thoughtfully and clearly written. I think the definition of R = R_3/R_2 is a novel and useful metric for evaluating TTIQ. I also like the interactive graphic. Meanwhile, please address all comments made by the reviewers. Reviewer 1's insightful comments should help to enrich the paper. However per PLOS One's publication criteria it is not necessary to pursue additional analyses - provided the model is clearly articulated and limitations are identified. In addition, I like Reviewer 2's suggestion to make code available and to include a paragraph that relates the methodological results to specific public health interventions. Please note that Reviewer 2's listed references on transmission heterogeneity do not need to be included.

We look forward to receiving your revised manuscript.

Kind regards,

Seth Blumberg

Academic Editor

PLOS ONE

Journal Requirements:

"This study was funded by the Swiss National Science Foundation (grant no. 310030B_-176401)."

"This study was funded by the Swiss National Science Foundation (grant no. 310030B_176401, awarded to SB). The funders had no role in study design, data collection and analysis, decision to publish, or preparation of the manuscript."

Reviewers' comments:

Reviewer's Responses to Questions

**Comments to the Author**

1. Is the manuscript technically sound, and do the data support the conclusions?

Reviewer #1: Yes

Reviewer #2: Yes

2. Has the statistical analysis been performed appropriately and rigorously? 

Reviewer #1: Yes

Reviewer #2: Yes

3. Have the authors made all data underlying the findings in their manuscript fully available?

Reviewer #1: Yes

Reviewer #2: No

4. Is the manuscript presented in an intelligible fashion and written in standard English?

Reviewer #1: Yes

Reviewer #2: Yes

5. Review Comments to the Author

Reviewer #1: ### Summary

In their manuscript, Ashcroft and coauthors propose a branching process model to assess the effectiveness of test-trace-isolate-quarantine TTIQ interventions on the containment of COVID-19. Their findings are overall consistent with the large body of evidence showing that TTI may help curb the spread of infectious diseases --- if done properly [1--6]. However, while the authors incorporate great detail into the transmission of COVID-19 by using empirical distributions for the generation/serial intervals and the time from contagion to symptoms onset, imperfections of the TTIQ interventions and all connection with its real-life implementation and its challenges are overseen (e.g., imperfect isolation, limited contact tracing capacity, the cost-effectiveness of quarantining large fractions of the population [2--4,6]).

Below find some observations for improvement.

## Major

1. Currently, there is no category for recovered/vaccinated individuals --- how does epidemic spread affect the baseline reproduction number? How do the authors compute current COVID-19 incidence?

2. What kind of contact tracing is here considered? If manual, there should be a maximum incidence from which no more contacts could be treated or only primary contacts would be prioritized (e.g., $\\tau\\to 0$ in the notation used in the manuscript) [3,4]. If digital contact tracing was used, how is the threshold defined? What's the cost of quarantine and the fraction of the population currently isolated? see, e.g., [6]

3. The definition of _symptoms_ is ambiguous; do the authors refer to COVID-specific symptoms, like loss of smell/taste or other less common symptoms? [7] This could affect the value of $a$ and $\\alpha$. How do symptoms-specificity relate to testing criteria in the event of high COVID-19 incidence?

4. How does the baseline reproduction number relate to the data-driven, testing-dependent observed reproduction number? how does the latter relate to $R_0$, $R_{\\text{TTIQ}}$, and $R_{\\rm TI}$? All reproduction numbers should share the tipping point at $R=1$, but their absolute value depends on the assumptions of the generation interval.

5. Despite being conceptually different, isolation and quarantine are implemented in the same way in the model (i.e., individuals are removed from the infectious pool and do not contribute further to spread). Isolation/Quarantine is deemed imperfect, as individuals share households, choose not to comply with the instructions, or hide their diagnosis because of economic pressure. The above would lead to identified (suspected) and unidentified new infections, contributing further to the spread.

6. What are the testing rates and absolute values behind $f$ and $g$? for $f$, the sensitivity of self-administrated tests is considerably lower than point-of-care administrated tests [8], and thus higher rates of false negatives would arise. On the other hand, point-of-care tests are limited, and individuals showing symptoms are told not to go there, thus also favoring underreporting. For a given value of $f$, how many tests per million per day have to be administrated? And for a given value of $g$ and an average number of close contacts per index case, how many calls have to be performed by the tracing agencies before finding a $g$ % of the newly generated cases? Is it reasonable to assume that all of them would be found exactly $\\Delta_2$ days after the report? How does the model deal with individuals that are simultaneously identified as close contact and index case?

7. I wonder whether it is correct to compare the efficacy of testing-isolating and tracing-quarantining in absolute terms. As I understand it, contact tracing cannot happen without testing; thus, it is conditional to it.

8. There is something odd with Figure S6; I would have expected the trends to be the other way around (if analyzed conditional one to each other). Currently, it seems that to improve the efficacy of contact tracing, we would have to miss more cases in the testing stage.

## Minor

1. Abstract and throughout the manuscript: currently, it reads "SARS-CoV-2 pandemic", but it should be "COVID-19 pandemic", as the latter refers to the disease.

2. There are parts in the introduction that would better fit (and are redundant with) the discussions section.

3. Lines 126--128: missing reference?

4. Figures in general: perhaps larger tick labels and larger fonts for figure legends

5. Discussion: (e.g., lines 418--422) Sensitivity analysis is typically included in the supplementary materials, as in the study of Kretzschmar and coauthors [9].

6. A sensitivity analysis for the asymptomatic fraction should also be performed (as it depends on testing criteria and it's likely to impact the effectiveness of testing policies)

7. How would random testing be implemented in this framework?

8. Currently, delays are modeled as a fixed parameter. However, how late an individual receives a positive test result is a random variable likely to be overdispersed (and the same for contact tracing). Can the authors perhaps discuss how this would affect their results?

## References

[1] Kerr, C. C., Mistry, D., Stuart, R. M., Rosenfeld, K., Hart, G. R., Núñez, R. C., ... & Klein, D. J. (2021). Controlling COVID-19 via test-trace-quarantine. Nature communications, 12(1), 1-12.

[2] Kretzschmar, M. E., Rozhnova, G., & van Boven, M. (2021). Isolation and contact tracing can tip the scale to containment of COVID-19 in populations with social distancing. Frontiers in Physics, 8, 677.

[3] Contreras, S., Dehning, J., Loidolt, M., Zierenberg, J., Spitzner, F. P., Urrea-Quintero, J. H., ... & Priesemann, V. (2021). The challenges of containing SARS-CoV-2 via test-trace-and-isolate. Nature communications, 12(1), 1-13.

[4] Contreras, S., Dehning, J., Mohr, S. B., Bauer, S., Spitzner, F. P., & Priesemann, V. (2020). Low case numbers enable long-term stable pandemic control without lockdowns. arXiv preprint arXiv:2011.11413.

[5] Gardner, B. J., & Kilpatrick, A. M. (2021). Contact tracing efficiency, transmission heterogeneity, and accelerating COVID-19 epidemics. PLOS Computational Biology, 17(6), e1009122.

[6] Lunz, D., Batt, G., & Ruess, J. (2021). To quarantine, or not to quarantine: A theoretical framework for disease control via contact tracing. Epidemics, 34, 100428.

[7] Nasserie, T., Hittle, M., & Goodman, S. N. (2021). Assessment of the Frequency and Variety of Persistent Symptoms Among Patients With COVID-19: A Systematic Review. JAMA network open, 4(5), e2111417-e2111417.

[8] Lindner, A. K., Nikolai, O., Kausch, F., Wintel, M., Hommes, F., Gertler, M., ... & Denkinger, C. M. (2021). Head-to-head comparison of SARS-CoV-2 antigen-detecting rapid test with self-collected nasal swab versus professional-collected nasopharyngeal swab. European Respiratory Journal, 57(4).

[9] Kretzschmar, Mirjam E., et al. "Impact of delays on effectiveness of contact tracing strategies for COVID-19: a modelling study." The Lancet Public Health 5.8 (2020): e452-e459.

Reviewer #2: Summary:

The authors present a well-written and devised study investigating the effect of test-trace-isolate-quarantine (TTIQ) strategies on SARS-CoV2 transmission. Empirical distributions of the generation time, infectivity profile, and incubation period are incorporated into a branching process model with parameters affecting reductions in the distribution of infectivity through time. Early isolation of index cases is found to be the most effective TTIQ strategy and the authors communicate uncertainty in the results exceptionally well throughout. There are a number of typos and a couple aspects of the methods that are a bit unclear. In addition, further discussion or incorporation of the effects of individual heterogeneities in R should be incorporated. Finally, making the code accessible in addition to the app would be helpful for transparency/reproducibility and future derivative work.

Minor Comments & Suggestions:

• Line 39: Introduce TTIQ acronym first time appearing in summary

• Line 64: “were” rather than “are”

• Line 79: “Testing and quarantine do not…” rather than “Testing and tracing does not…”

• Line 124: Might be worth explaining how this is a methodological advance over Fraser et al (2004) by calling attention to a few specific details

• Line 145: Seems like the definition of the infectivity profile should include reference to symptomatic vs asymptomatic infection and clarify how it is defined in the case of an asymptomatic infection. Looks like it’s mentioned in the discussion, but probably worth stating in the methods

• Figure 1 is quite nice, but it’s a bit unclear what the y-axis is meant to represent. Is it the probability of generating a new case at time t? Such that infecteds are more likely to generate new cases around the time of their symptom onset? This makes sense, even as it’s not explicitly incorporated into the distributions used to generate Fig 1, but it’s worth a bit more explicit discussion. Also, in the event that the y axis does represent this transmission probability, it might be worth incorporating this into the timing of secondary cases in the figure such that more of the secondary cases are generated around the time of highest infectiousness, but this is just a minor suggestion that might not be worth the effort to re-configure the figure.

• Code and details such as coding language and additional software packages used should be made available in addition to the app

• Figure 2A: consider using color-blind friendly color palate

• Not sure if possible as I’m not familiar with LDA, but would be very interesting to perform the same analysis on pairs of parameters, i.e. could answer: f is most impactful, but what is the most impactful parameter that interacts with f?

• Line 394: “than” rather than “that”

• Line 452: “contacts” rather than “contracts”

• Might be worth adding a paragraph in the discussion suggesting ways to enact the most impactful TTIQ interventions. This would help translate the results into actionable policy for public health practitioners that may not fully understand the methods and approach. Widely available rapid testing for instance could be suggested as a way to increase f.

o Larremore, D. B., Wilder, B., Lester, E., Shehata, S., Burke, J. M., Hay, J. A., ... & Parker, R. (2021). Test sensitivity is secondary to frequency and turnaround time for COVID-19 screening. Science advances, 7(1), eabd5393.

Major comments:

• Given the theoretical foundation in branching process theory, it’s worth investigating the implications of the variance of R on the results in addition to its mean. Previous theoretical work on superspreading and the influence of the dispersion parameter assuming R is negative binomially distributed have found it’s important. Especially given the impact of f (fraction of index cases identified), this could lead to interesting insights. If nothing else, I think it’s worth a paragraph in the discussion, as my intuition is that it would affect the variance/confidence intervals of the results and not so much the mean.

o Blumberg, S., & Lloyd-Smith, J. O. (2013). Comparing methods for estimating R0 from the size distribution of subcritical transmission chains. Epidemics, 5(3), 131-145.

o Blumberg, S., & Lloyd-Smith, J. O. (2013). Inference of R 0 and transmission heterogeneity from the size distribution of stuttering chains. PLoS computational biology, 9(5), e1002993.

o Lloyd-Smith, J. O., Schreiber, S. J., Kopp, P. E., & Getz, W. M. (2005). Superspreading and the effect of individual variation on disease emergence. Nature, 438(7066), 355-359.

6. PLOS authors have the option to publish the peer review history of their article (what does this mean?). If published, this will include your full peer review and any attached files.

Reviewer #1: No

Reviewer #2: No

---

## [Decision Letter · Decision Letter 1]

24 Jan 2022

Test-trace-isolate-quarantine (TTIQ) intervention strategies after symptomatic COVID-19 case identification

PONE-D-21-26262R1

Dear Dr. Ashcroft,

We’re pleased to inform you that your manuscript has been judged scientifically suitable for publication and will be formally accepted for publication once it meets all outstanding technical requirements.

Kind regards,

Seth Blumberg

Academic Editor

PLOS ONE

Additional Editor Comments (optional):

Thank you for your diligent attention the reviewer comments. I am impressed with this careful, thorough and interesting research.

Reviewers' comments:

Reviewer's Responses to Questions

**Comments to the Author**

1. If the authors have adequately addressed your comments raised in a previous round of review and you feel that this manuscript is now acceptable for publication, you may indicate that here to bypass the “Comments to the Author” section, enter your conflict of interest statement in the “Confidential to Editor” section, and submit your "Accept" recommendation.

Reviewer #1: All comments have been addressed

2. Is the manuscript technically sound, and do the data support the conclusions?

Reviewer #1: (No Response)

3. Has the statistical analysis been performed appropriately and rigorously? 

Reviewer #1: (No Response)

4. Have the authors made all data underlying the findings in their manuscript fully available?

Reviewer #1: (No Response)

5. Is the manuscript presented in an intelligible fashion and written in standard English?

Reviewer #1: (No Response)

6. Review Comments to the Author

Reviewer #1: I thank the authors for the careful consideration of my comments and suggestions. In my opinion, the revised version constitutes a solid contribution to the state-of-the-art, and I congratulate the authors on the fruit of their research.

7. PLOS authors have the option to publish the peer review history of their article (what does this mean?). If published, this will include your full peer review and any attached files.

Reviewer #1: No

---

## [Editor Report · Acceptance letter]

3 Feb 2022

PONE-D-21-26262R1 

Test-trace-isolate-quarantine (TTIQ) intervention strategies after symptomatic COVID-19 case identification 

Dear Dr. Ashcroft:

I'm pleased to inform you that your manuscript has been deemed suitable for publication in PLOS ONE. Congratulations! Your manuscript is now with our production department. 

Kind regards, 

on behalf of

Dr. Seth Blumberg 

Academic Editor

PLOS ONE